# Improved Pedestrian Dead Reckoning Based on a Robust Adaptive Kalman Filter for Indoor Inertial Location System

**DOI:** 10.3390/s19020294

**Published:** 2019-01-12

**Authors:** Qigao Fan, Hai Zhang, Peng Pan, Xiangpeng Zhuang, Jie Jia, Pengsong Zhang, Zhengqing Zhao, Gaowen Zhu, Yuanyuan Tang

**Affiliations:** 1Internet of Things Engineering, Jiangnan University, Wuxi 214000, China; qgfan@jiangnan.edu.cn (Q.F.); 6161920011@vip.jiangnan.edu.cn (X.Z.); 6171920006@stu.jiangnan.edu.cn (J.J.); 1070114134@vip.jiangnan.edu.cn (P.Z.); 6181915019@stu.jiangnan.edu.cn (Z.Z.); 6181920012@stu.jiangnan.edu.cn (G.Z.); 6181915013@stu.jiangnan.edu.cn (Y.T.); 2Department of Mechanical Engineering, McGill University, Montreal, QC H3A 0G4, Canada; peng.pan@mail.mcgill.ca

**Keywords:** indoor inertial positioning, MEMS-IMU, improved pedestrian dead reckoning, robust adaptive Kalman filter

## Abstract

Pedestrian dead reckoning (PDR) systems based on a microelectromechanical-inertial measurement unit (MEMS-IMU) providing advantages of full autonomy and strong anti-jamming performance are becoming a feasible choice for pedestrian indoor positioning. In order to realize the accurate positioning of pedestrians in a closed environment, an improved pedestrian dead reckoning algorithm, mainly including improved step estimation and heading estimation, is proposed in this paper. Firstly, the original signal is preprocessed using the wavelet denoising algorithm. Then, the multi-threshold method is proposed to ameliorate the step estimation algorithm. For heading estimation suffering from accumulated error and outliers, robust adaptive Kalman filter (RAKF) algorithm is proposed in this paper, and combined with complementary filter to improve positioning accuracy. Finally, an experimental platform with inertial sensors as the core is constructed. Experimental results show that positioning error is less than 2.5% of the total distance, which is ideal for accurate positioning of pedestrians in enclosed environment.

## 1. Introduction

Indoor positioning is a technology for positioning in an indoor environment, which shows great prospects in industries such as emergency rescue, logistics, wireless games and shopping malls [1,2,3]. The Global Navigation Satellite System, such as the Global Positioning System (GPS), BeiDou Navigation Satellite System (BDS), and Global Navigation Satellite System (GLONASS), can provide absolute accurate positioning, but its performance in some enclosed environments, such as tunnels and mines, is not satisfied; as the Global Navigation Satellite System may suffer heavily from satellite signal blockage [4]. Therefore, how to obtain accurate position information of indoor objects has attracted wide attention. At present, indoor positioning methods have been proposed, including Ultra Wideband (UWB) [5], ZigBee [6], Wi-Fi [7], Bluetooth [8], Radio Frequency Identification (RFID) [9], and visual positioning [10]. However, performances of these positioning methods are easily affected by external environment. UWB positioning is susceptible to non-line-of-sight interference such as occlusion, external collision and strong magnetic field interference [11]. Wi-Fi positioning is susceptible to interference from other signals and consumes a lot of power. In addition, ZigBee and Bluetooth positioning are not suitable for complex environments, in which the performance is not stable. RFID positioning does not have communication capabilities, and the anti-interference ability is poor. Visual positioning has poor real-time performance and is limited by light conditions, which cannot work in dark environments [12]. With the continual miniaturization of Micro-Electro-Mechanical Systems (MEMS) in recent years, the MEMS-IMU has the advantages of simple structure and portability. At the same time, inertial measurement is completely autonomous, suitable for all weather, free from external environment interference, and has no signal loss, so MEMS-IMU-based inertial navigation technology is very suitable for indoor positioning [13,14].

Pedestrian dead reckoning (PDR) is an effective technique for indoor positioning of pedestrians [15,16,17]. By measuring the movement information of pedestrians in real time, the pedestrian’s step, step length, and heading angle are identified to calculate the position of the pedestrian. The indoor positioning based on MEMS-IMU is widely used because of its advantages such as portability, simple structure, and strong anti-interference performance. However, errors of the Inertial positioning system accumulated over time, which greatly affects the positioning accuracy [18]. Domestic and foreign experts have proposed combined positioning system to suppress cumulative errors through auxiliary positioning techniques, such as INS/GNSS [19], INS/RFID [20], and INS/UWB positioning [21]. The combined positioning technology can improve the positioning accuracy compared with the single inertial positioning technology, but its external requirements are more demanding, which greatly limits the scope of use of the combined positioning system. In scenes with harsh external environments, such as the disaster scene of mines or tunnels, the combined positioning system is difficult to build and wastes valuable rescue time. The single inertial positioning technology is extremely suitable because of its characteristics, so the study of single inertial positioning system has high practical value.

In the framework of pedestrian dead reckoning, the step estimation is very important. The key to correctly estimate step frequency is to avoid acceleration warpage and sensor noise, which may lead to incorrect step estimation. Noises usually exit in the signal obtained by MEMS-IMU, which result in the poor positioning result. To improve accuracy of positioning, it is necessary to preprocess the signal. Low pass filter is typically used to smooth the data, but it is difficult to determine the appropriate cutoff frequency [22]. The authors in [23] analyze error noise term by Allan variance method, and optimized Kalman filter is designed to denoise the signal according to noise correlation, but the amount of calculation is large. At present, the step estimation algorithm usually includes Peak detection [24,25,26], Zero crossings [27] and fast Fourier transform [28], where the signals of accelerometer are generally used. In these algorithms, the placement of the IMU defaults to one of the accelerometer axes being always vertical [29], which greatly affects the shape of the accelerometer signal and the effectiveness of the algorithm.

Step length estimation plays an important role in the pedestrian dead reckoning. For different individuals, the step length of different individuals varies. Even for the same person, step length will change due to different states of motion. Usually, there are two kinds of models for estimating the step length: static model and dynamic model. It is assumed that in the static model, the step length is a fixed value, which is obviously inaccurate. In comparison, the dynamic model considers that any effective stride has a different step length, which is affected by the different individuals or different states of motion. The step dynamic model includes step-frequency linear model [30], nonlinear model [31], and adaptive learning step length model. A nonlinear model is used in [32] to accommodate the difference of step length between different pedestrians, which highly stable and moderately complex compared to other dynamic step models.

Heading estimation is the core of the pedestrian dead reckoning algorithm. Calculating the heading angle using the magnetometer has the advantages of simple structure, low power consumption, and no cumulative errors [33]. However, magnetometers are susceptible to external environmental disturbances, especially in mines or garages, where there are many ferromagnetic materials. There are serious cumulative errors when using the gyroscope alone, especially at corners. The longer the time is, the larger the cumulative error will be [34]. In order to improve the heading accuracy of the inertial measurement unit, fusion filtering algorithms are proposed, such as Kalman filtering [35] and complementary filtering [36]. The authors in [37] use an extended Kalman filter based on two-level quaternions to fuse signals of acceleration, angular velocity and magnetic to minimize the cumulative error, but ignores the change in noise characteristics. Therefore, the adaptive filtering algorithm is proposed. In the literature [38], the fuzzy adaptive Kalman filter is used to fuse the signals of gyroscope and accelerometer, but it is limited by the precision of the fuzzy control, and the calculation amount is large. Sage–Husa adaptive filtering [39] can correct the statistical characteristics of process noise and measurement noise in real time to improve the filtering accuracy, but it is difficult to give accurate noise characteristics and the calculation amount is large. In comparison, the complementary fusion filter [40] complements the signals of accelerometer, magnetometer and gyroscope with a small amount of calculation, but high precision. The paper is structured as follows: Section 2 briefly introduces the indoor positioning model based on the PDR algorithm firstly, and then the wavelet decomposition is used to preprocess the original signal, finally the proposed improved PDR algorithm is discussed in detail, including improved step estimation, length estimation, and heading estimation based on fusion filtering. In Section 3, several experiments are performed and the experimental results are analyzed based on established experimental platform. Some final remakes in Section 4 end the paper.

## 2. Materials and Methods

### 2.1. System Modeling

#### 2.1.1. Pedestrian Dead Reckoning

Pedestrian dead reckoning algorithm is an effective way to achieve pedestrian positioning in a closed environment. In PDR algorithm, the step frequency and length of penetration are estimated based on information of acceleration, respectively. The pedestrian’s movement direction can be determined from the gyroscope’s output. Thus the position of pedestrian can be calculated based on the step, step length and heading. The pedestrian positioning model based on PDR is expressed as follows:

In Figure 1, black marks represent footprint of pedestrian, and arrows indicate the direction of pedestrian movement. *L_k_* and *φ_k_* represent pedestrian’s step length and heading at step *k*. (*x_k_*, *y_k_*) represents the pedestrian position at step *k*. Pedestrian positioning equation is shown as follows:(1)$$\begin{array}{l} x_{k+1}=x_{k}+L_{k}\sin\varphi_{k}\\ y_{k+1}=y_{k}+L_{k}\cos\varphi_{k} \end{array}$$

#### 2.1.2. Pedestrian Positioning System Model

Pedestrian indoor positioning model describes the whole process from data acquisition to position output, as shown in Figure 2. In this model, MEMS-IMU part integrates several inertial sensors such as accelerometer, magnetometer, and gyroscope. The raw data of acceleration, angular velocity and magnetic field strength acquired through the MEMS-IMU suffers seriously from low signal to noise ratio. Therefore, it is necessary to preprocess the raw signal by wavelet denoising. After preprocessing, the information of acceleration is used to estimate steps and step length. Accelerometer and gyroscope complement each other to get the initial attitude angle by complementary filter, which then are taken as measured value of robust adaptive Kalman filter (RAKF). To suppress the interference of outliers, RAKF is proposed based on adaptive Kalman filter, and optimal heading angle *φ* is obtained. According to the information of step *n*, length *l* and heading angle, the pedestrian’s position is calculated based on Equation (1).

### 2.2. Signal Preprocessing

The MEMS inertial system is widely used in positioning systems because of its low cost, light weight, and complete autonomy. However, the inertial sensor is easily affected by the determination error and the random error. The determination error is closely related to the MEMS structure and the processing technology, and reducing this error is expensive and difficult. Therefore, random error’s processing of inertial sensors becomes the focus of MEMS data processing [41]. Typical denoising methods include digital filtering and wavelet denoising. Digital filtering separates the wanted signal and noise to denoise without overlapping the spectrum of the signal and the spectrum of the noise. The spectrum of the actual situation signal and the spectrum of the noise tend to overlap. For example, the spectrum of Gaussian white noise is distributed almost in the entire frequency domain. Wavelet denoising can be used to well separate random noise. Therefore, the original signal is preprocessed by wavelet denoising.

As an effective signal digital processing method, wavelet denoising is based on the different characteristics of signals and noise at various scales. The wavelet coefficients are obtained by following certain threshold criteria to achieve effective denoising. Wavelet denoising can be divided into the following three steps:(1)Wavelet decomposition of signals containing noise: select the appropriate wavelet base and decomposition layer;(2)Threshold quantization processing: select appropriate thresholds and threshold functions to process the coefficients of each layer;(3)Wavelet reconstruction: reconstruct the processed coefficients to obtain the denoised signal.

In the process of wavelet denoising, the choice of threshold is very important, usually using hard threshold and soft threshold method. The hard threshold method is prone to edge blurring and Gibbs phenomenon; the soft threshold method changes the degree of approximation between the reconstructed signal and the original signal. The improved threshold function is proposed from the literature [42]:(2)$$T(u,\lambda_{1},\lambda_{2},m)=\{\begin{array}{c} u-\frac{1}{2}\mathrm{sgn}(u)\frac{\lambda_{2}^{m}}{u^{m-1}},\left|u\right|\geq \lambda_{2}\\ \frac{1}{2}\mathrm{sgn}(u)\frac{\lambda_{2}{\left(\left|u\right|-\lambda_{1}\right)}^{m}}{{\left(\lambda_{2}-\lambda_{1}\right)}^{m-1}},\lambda_{1}\leq \left|u\right|\leq \lambda_{2}\\ 0,\left|u\right|\leq \lambda_{1} \end{array}$$
where *T* is the denoising operation, that is, by the threshold and the threshold function, filtering out the noise coefficient and retaining the detail coefficient. *u* is the wavelet coefficient before denoising, and *λ*_1_ and *λ*_2_ are the lower threshold and the upper threshold. The adjustment parameter m can change the lower threshold so that the improved threshold function tends to be between the soft threshold function and the hard threshold function, having both the advantages of them.

### 2.3. Pedestrian Dead Reckoning

Pedestrian dead reckoning uses the inertial measurement unit to calculate the position of the next moment based on current position. It has advantages of simple structure, small size, full autonomy, and strong anti-interference characteristics. The algorithm includes three important parts: step estimation, length estimation, and heading estimation.

#### 2.3.1. Step Estimation

Acceleration has periodicity during normal pedestrian motion, and peak detection is an effective means to determine the pedestrian’s step. In order to avoid the influence of system errors and the specific orientation of the accelerometer, the total acceleration *A* of the three axes is calculated:(3)$$A=\sqrt{A_{x}{}^{2}+A_{y}{}^{2}+A_{z}{}^{2}}$$
where *A_x_*, *A_y_*, *A_z_* represent the corresponding acceleration of three axes of accelerometer.

The key to peak detection is to avoid interference from abnormal peaks, and conventional peak detection uses a fixed time window to constrain the peaks, which ignores the constraints of adjacent peaks on acceleration. In this paper, the multi-threshold is used to constrain the acceleration peak to improve the accuracy of the step estimation. A set of constraints (*p*, *v*, Δ*p_t_*, Δ*v_t_*, |Δ*p*|, |Δ*v*|) represent peaks, valleys, time differences of adjacent peak, time differences of adjacent valley, differences of adjacent peak-to-peak, and differences of adjacent valley-to-valley. The effective peak and valley discriminant equations are shown as follows:(4)$$peak=\{\begin{array}{c} 1,\;10<p_{\eta }<p\&△p_{t\eta -}<△p_{t}<△p_{t\eta +}\&0<\left|△p\right|<△p_{\eta }\\ 0,others \end{array}$$
wherein peak = 1 indicates it is an effective peak; and peak = 0 indicates a pseudo peak. (*p_η_*, Δ*p_tη−_*, Δ*p_tη+_*, Δ*p_η_*) is the threshold group for peak detection.
(5)$$valley=\{\begin{array}{c} 1,\;v<v_{\eta }<10\&△v_{t\eta -}<△v_{t}<△v_{t\eta +}\&0<\left|△v\right|<△v_{\eta }\\ 0,others \end{array}$$
wherein valley = 1 indicates it is an effective valley; and valley = 0 indicates a pseudo valley. (*v_η_*, Δ*v_tη_*_−_, Δ*v_tη+_*, Δ*v_η_*) is the threshold group for valley detection. When stationary, the accelerometer is affected by the acceleration of gravity, and the total acceleration *A* is approximately equal to 10. The cycle of foot movement starts with the footstep off the ground and ends with the footstep touching the ground again. At the start, the upward acceleration is opposite to the direction of gravity acceleration, and the direction is the same when re-contacting, so set *p_η_* > 10 and *v**_η_* < 10. An effective peak and a valley adjacent to each other constitute an effective step.

In summary, the step detection based on the triple constraint condition considers the constraint of the adjacent peaks on the pedestrian acceleration, which can avoid the misjudgment problem of the traditional PDR gait detection method, and thus realize the accurate judgment of the step number.

#### 2.3.2. Length Estimation

A precise accurate dynamic length estimation model allows researchers to improve the accuracy of pedestrian dead reckoning. The authors in [43] use the nonlinear step size model to estimate the step length, and the improved model is in this paper is effective for step length estimation of different pedestrian individuals.
(6)$$L_{t}=\alpha \sqrt[{4}]{A_{\max}-A_{\min}}+\beta (A_{\max}-A_{\min})$$

As known to all, different individuals have different parameters α and β. To enable this model suitable for different individual, α and β should be obtained by performing the least-multiplication fitting of the training data before positioning.

#### 2.3.3. Heading Estimation

The heading estimate is a crucial part of the pedestrian dead reckoning algorithm, which is calculated from measurement data of strapdown MEMS-IMU. A data fusion algorithm of attitude angle can be roughly divided into two categories. One is based on the frequency domain characteristics, the typical algorithm of which is complementary filter. This fusion algorithm does not need to consider the statistical characteristics of the signal and noise. The other is to design filters in the time domain using the state-space approach, such as the Kalman filter. As complementary filters generally do not require an accurate noise model, it has advantages of less computation and higher accuracy. However, complementary filters cannot suppress the measurement of outliers. Therefore, this paper proposes a fusion filter algorithm consisting of complementary filter and robust adaptive Kalman filter (RAKF). Take pitch θ as an example, and the fusion filter model is shown as follows:

(1) Complementary Filter

MEMS inertial sensors have some inherent problems. There is a static drift in the gyroscope, and errors will accumulate when the attitude is calculated; the dynamic response of the accelerometer and magnetometer is poor, but there is no cumulative error. Therefore, the attitude of accelerometer and magnetometer, the attitude of gyroscope are calculated under static and dynamic conditions, respectively, and then the outputs are complementary to each other to improve the accuracy of the attitude. Complementary filter algorithm is shown in Figure 3 in details.
(7)$$\begin{array}{c} \;\theta_{err}=\theta_{a}-\theta \\ \;\theta_{a}^{\prime }=K_{p}\theta_{err}+\int_{0}^{t}K_{i}\theta_{err}\\ \theta =\frac{\tau }{\tau +dt}\theta_{g}+\frac{dt}{\tau +dt}\theta_{a}^{\prime } \end{array}$$

Because the low-passband attenuation of the complementary filter is slow, and the error is large when the noise is large, the Proportional Integral(PI) control is added on the basis of the complementary filter. The error between the attitude angle calculated by accelerometer and magnetometer, and the post-filtering attitude angle are PI-controlled. Then the complementary filter algorithm is used to fuse the attitude information and improve the accuracy of the attitude. In Figure 3, *τ*/*τ* + *dt* is high-pass filter and *dt*/*τ* + *dt* is low-pass filter. The output of the accelerometer and magnetometer passes through a low-pass filter to limit the high-frequency jitter in the attitude measurement; accumulated drift error of the gyroscope can be suppressed through the high-pass filter. The parameters are selected according to reference [44]. *K_p_* is the switching frequency of the low-pass filter and the high-pass filter, which is selected according to the frequency characteristics of the accelerometer and the output of the gyroscope, and *K_p_* = *K_i_*^2^/2. The parameters *K_p_*, *K_i_* of the complementary filter in this paper are set to 0.7 and 1.2 respectively, where the error correction effect is obvious.

(2) Robust Adaptive Kalman Filter

Complementary filter complements accelerometer, magnetometer with gyroscope to suppress cumulative errors, which improves the accuracy of heading. However, the complementary algorithm is limited by the inaccurate measurement of the sensors. The outliers in the measured value cannot be identified and the interference cannot be suppressed, which severely reduces the accuracy and affects stability of the positioning system. On the basis of a complementary filter, the robust adaptive Kalman filter algorithm is proposed in this paper. The post-complementary filtered attitude and Optimal attitude are measured value and state value, respectively, to suppress the influence of the measured outliers on the heading estimation.

The basic idea of Kalman filter is that based on the state model of signal and noise, the estimate of the current state variable is updated using the previous time’s estimated value and the current time’s observation value. The equation of state for the pedestrian positioning system is described below:(8)$$x(k+1)=x(k)+\left[\begin{array}{ccc} 1 & \sin\phi \prime_{k} & \cos\phi \prime_{k}\tan\theta \prime_{k}\\ 0 & \cos\phi \prime_{k} & -\sin\phi \prime_{k}\\ 0 & \sin\phi \prime_{k}\sec\theta \prime_{k} & \cos\phi \prime_{k}\sec\theta \prime_{k} \end{array}\right]\left[\begin{array}{c} w_{xk}\\ w_{yk}\\ w_{zk} \end{array}\right]+w(k)$$
where *x*(*k*) = [*Φ*′(*k*), *θ*′(*k*), *ψ*′(*k*)] is the state variable, *w*(*k*) represents system noise vector, which is ignored in this stable positioning system.

The complementary filtered attitude angles are selected as the measured value *Z*(k) = [*Φ*(*k*), *θ*(*k*), *ψ*(*k*)], the observation equation is shown as follow:(9)Z(k+1)=H(k+1)x(k)+v(k+1)
where *H*(*k*) = diag [1, 1, 1], and *v*(*k* + 1) is the observation noise vector.

The Kalman filter consists of a prediction process and an update process. The calculation steps are as follows:

(1) Forecasting process:

State forecast
(10)$${\hat{X}}_{k,k-1}=\Phi_{k,k-1}{\hat{X}}_{k-1}$$

Covariance matrix of state prediction
(11)$$P_{k,k-1}=\Phi_{k,k-1}P_{k-1}\Phi_{k,k-1}^{T}+Q_{k-1}$$

(2) Updating process:

Gain matrix
(12)$$K_{k}=P_{k,k-1}H_{k}^{T}{\left[H_{k}P_{k,k-1}H_{k}^{T}+R_{k}\right]}^{-1}$$

State estimation
(13)$${\hat{X}}_{k}={\hat{X}}_{k,k-1}+K_{k}\left[Z_{k}-H_{k}{\hat{X}}_{k,k-1}\right]$$

Covariance matrix of state estimation
(14)$$P_{k}=\left[I-K_{k}H_{k}\right]P_{k,k-1}{\left[I-K_{k}H_{k}\right]}^{T}+K_{k}R_{k-1}K_{k}^{T}$$

By adding a measurement noise estimator to the traditional Kalman [45], the statistical properties of the measurement noise can be estimated and corrected in real time.
(15)$$R_{k}=\left(1-d_{k}\right)R_{k-1}+d_{k}\left\{\left[I-H_{k}K_{k-1}\right]e_{k}e_{k}^{T}{\left[I-H_{k}K_{k-1}\right]}^{T}+H_{k}P_{k-1}H_{k}^{T}\right\}$$
(16)$$d_{k}=\frac{1-b}{\;1-b^{k+1}}$$
where in *b* is forgetting factor, and the range of *b* is 0.95–0.99 in reference [45], which is set to 0.98 in the paper; the definition innovation matrix *e_k_* and the corresponding covariance matrix *C_k_* at time *k* are:(17)$$e_{k}=Z_{k}-H_{k}{\hat{X}}_{k/k-1}$$
(18)$$C_{k}=H_{k}P_{k,k-1}H_{k}^{T}+R_{k}$$

When the system is not faulty, the innovation is a zero-mean white noise sequence. If the actual statistical characteristics of the innovation are not consistent with the theory, the process is considered to be abnormal. In terms of fault detection, the innovation *χ*^2^ detection algorithm is an effective method.

Assume test statistic
(19)$$T_{k}=e_{{}_{k}}^{T}C_{{}_{k}}^{-1}e_{k}$$

The innovation *e_k_* is a normal distribution with zero mean, so *T_k_* obeys the *χ*^2^ distribution with a degree of freedom *m* (*m* is the dimension of the observation). Transform the fault detection condition into a hypothesis test condition, according to the definition of the *χ*^2^ distribution:(20)$$\{\begin{array}{c} \;H_{0}:T_{k}∼\chi^{2}(m,0)\\ \;H_{1}:T_{k}∼\chi^{2}(m,\lambda ) \end{array}$$

In the formula, *H*_0_ indicates that there is no gross error in the observed value, *H*_1_ indicates that the observed value has an abnormal value, and *λ* is a non-centralized parameter. The significance level α indicates the probability that the null hypothesis is established but rejects the hypothesis selection alternative hypothesis. The selection of the significance level determines the threshold of the innovation statistic. The smaller the value of α, the larger the threshold, and too large or too small threshold will affect the system’s robust performance. A suitable value for the alpha should be in the range of 0.01 to 0.15, and we make α = 0.1 in the paper. The boundary condition is set to *T_D_* = *χ*^2^*_a_* (*m*, 0).

If *T_k_* ≤ *T_D_*, it is judged that there is no abnormality in the observation. Otherwise, there is an abnormal value in the observation. It is necessary to introduce a weighting factor to amplify the new covariance and reduce the influence of the wild value on the filtering. Construct weighting factor using Huber functions:(21)$$\alpha_{i}=\{\begin{array}{c} \;1:T_{k,i}\leq T_{D}\\ \;\frac{T_{D}}{T_{k,i}}:T_{k,i}≻T_{D} \end{array}(\alpha_{i}\leq 1)$$

And *a* = diag [*a*_1_, *a*_2_ … *a_n_*], *I* = 1, 2 … *n*. The equivalent covariance of the observations is updated to
(22)$$R_{k}=\frac{R_{k}}{\alpha }$$

It can be seen from Equation (12) that the adaptive innovation weighting factor (*a_i_* ≤ 1) reduces the gain matrix when the observed gross error occurs. The weight of the gross error in the state estimation is reduced, the interference of the wild value on the filtering is suppressed, and the fault tolerance of the system is improved. The whole RAKF algorithm is shown as below in details in Figure 4:

## 3. Results

### 3.1. Laboratory Equipment

The Mti series MEMS-IMU from Xsens is used in the experiment, which is fixed on the instep. The MEMS-IMU is mainly composed of a triaxle orthogonal accelerometer, a triaxle orthogonal magnetometer and a triaxle orthogonal gyroscope. Keep the foot unmoved for 1 s before the start of the experiment to coincide with the starting point. The MEMS-IMU used in the experiment meets the experimental requirements.

### 3.2. Analysis of Signal Preprocessing

In order to verify the validity of the wavelet decomposition, the angular velocity of the gyro *X*-axis in the static case is selected for analysis. In this paper, the wavelet decomposition selects ‘db6′ as the wavelet basis function according to reference [42]. The number of decomposition layers of 3 to 9 wavelets is selected for comparative study, and the optimal decomposition layer number of 9 is selected. The wavelet coefficients are processed by the improved threshold function. The parameters of improved threshold function are set according to the literature [42], where upper threshold *λ*_2_ is a fixed value 0.8; *m* is an integer from 2–13; and lower threshold *λ*_1_ is set to *λ*_2_/(*m* + 1). Draw a comparison of the original signal and the reconstructed colored noise, as shown in Figure 5. Compared with the original signal, the reconstructed signals are all closer to the true value, no matter what threshold functions are used, which proves the filtering effectiveness of the wavelet decomposition. Compared with traditional threshold functions, the improved threshold function in [42] perform better, which is adopted in this paper to improve the credibility of data.

### 3.3. Analysis of Improved PDR Algorithm

In order to verify the effectivity of the proposed improved PDR algorithm, step estimation, step length estimation, and heading estimation are tested separately before the formal experiment. The experimental site is arranged on the second floor of the entire School of Internet of Things Engineering. The satellite image of the experimental site is shown in Figure 6a. The structure and installation of the pedestrian inertial navigation system is shown in Figure 6b. A three-axis accelerometer, three-axis gyroscope, and three-axis magnetometer are mounted on the I2C bus and data are transmitted from the serial port to the host computer through the DSP. Acceleration, angular rate, and magnetic field strength measurements are obtained. The inertial navigation module is tied on the foot to obtain pedestrian movement information.

In Figure 6a, the red circle represents the starting point and the ending point. The direction of movement is indicated by the arrow. The area enclosed by the black dotted line is used to test the performance of the step estimation algorithm. The MEMS-IMU is fixed on the back of the pedestrian, and the subject moves along the arrow as shown in Figure 6b.

#### 3.3.1. Step Frequency Analysis

The multi-threshold method is used to ameliorate the step estimation algorithm in this paper, which can avoid interference from abnormal peaks. The threshold group (*p_η_*, Δ*p_tη−_*, Δ*p_tη+_*, Δ*p_η_*) for peak detection is set to (30, 0.6, 1.8, 20); and the threshold group for valley detection (v*η*, Δ*v_tη−_*, Δ*v_tη+_*, Δ*v_η_*) is set to (8, 0.6, 1.8, 5).

In order to verify the validity of the algorithm, the pedestrian moves at the above experimental site, and the acceleration information of the strap-down MEMS-IMU is obtained. Then, the wavelet denoising is adopted to improve reliability of acceleration data. Finally, the new data is processed to obtain the total acceleration of the three axes, and the multi-threshold method is used to discriminate the effective peaks and valleys, which form effective steps, as shown in Figure 7.

Figure 7a is the global processed acceleration waveform, and Figure 7b is a partial enlarged view of the first 15 s part. In Figure 7, it is obvious that A is approximately equal to 10 when stationary. According to the previously proposed step estimation algorithm, effective peaks and valleys are found as shown in Figure 7b, which are marked with red squares, and adjacent effective peak and valley can form an effective step. After multiple tests, the average detection accuracy is roughly 99%, which prove the stability and effectiveness of the step estimation algorithm.

#### 3.3.2. Step Length Analysis

The parameters α and β in Equation (6) vary from person to person due to factors such as height, weight, and gender, so different individuals must be trained to obtain the applicable parameters before positioning. In this paper, two subjects participated in length experiment, subject 1 is female, 165 cm height, 56 kg weight; subject 2 is male, 176 cm, and 72 kg. Two subjects are required to move at random to record training data as much as possible. Then the training data including step length and acceleration information is processed by the least squares method to obtain the corresponding parameters *α* and *β*. The parameters α and β of subjects are −0.6291 and 0.0868, −0.7291, and 0.0922, respectively.

For quantitative analysis, two subjects performed multiple tests in site shown in Figure 6b, including status of walking and running. Subjects move from red frame, along direction of arrows in a fixed step of 1 m (length of 2 tiles), and the realistic distance (RD) for length estimation is about 181m. Data is processed using the length estimation model in reference [43] and improved model of this paper, respectively. The step length and error of two subjects are intuitively compared in the Figure 8. Finally, the solution distance (SD) and mean absolute error (MAE) of length of both two subjects processed by different models are calculated. The results of same subject in the same state are averaged and shown in Table 1, where Run-1 represents the results of subject 1 under running.

In Figure 8, the left and right part represents the measured results in walking and running respectively. Through observing each single step, the length estimation accuracy of improved model is generally a little better than the original model, although not very obvious. In qualitative analysis of both two models, experimental data of the entire path, which contains both subjects is used to calculate mean absolute error and Solution distance. As shown in Table 1, the improved model performs better than the original model, regardless of subjects or motions. The accuracy of length estimation of run-2 is relatively poor, but the accuracy is still higher than 94%. The improved model can provide accurate length estimation in PDR algorithm, where parameters α and β need to be determined by individual in advance.

Different individuals need corresponding parameters. In order to highlight the importance of the corresponding parameters, the parameters of subject 1 are used to solve subject 2, and the results under two different motions are both shown as follows: the SD and MAE of subject 2 under running are 238.2 and 0.32, respectively. In the walking state, the solution distance and mean absolute error are 231.8 and 0.27, respectively. Compare the solution results of subject 2 using two different parameters, it is obvious that the results using the corresponding parameters, no matter original or improved model is used, are much better than the results under mismatched parameters. Therefore, it is necessary that different individuals should obtain the matching parameters before performing the positioning experiments.

#### 3.3.3. Heading Analysis

The improved heading estimation is the core of the proposed PDR algorithm, inheriting the advantages of complementary filtering (CF) and robust adaptive Kalman filter. In the PDR algorithm, subtle heading deviation will cause pedestrian to deviate significantly from the real position because of the cumulative effect, which severely affects the accuracy of the inertial positioning system. The heading tests are carried out in the above environment, and Figure 9 shows comparisons of performance of different algorithms in terms of heading error.

As shown in Figure 9, the performance of complementary filtering is not good enough, whose MAE is 18.6°. Compared to complementary filtering, CF + KF performs better, but still suffers from interference of abnormal values. In contrast, fusion filter introduces RAKF with complementary filtering, which can effectively reduce the influence of outliers on filtering to improve the accuracy of PDR positioning system. As shown in Figure 9b, the dynamic error of complementary filter and CF + KF are −58.6–12.8° and −10.8–39.8, showing that there are individual outliers. In comparison, the heading dynamic error of the fusion filter is −18.1–9.8, indicating that individual outliers are effectively suppressed.

#### 3.3.4. Trajectory Comparison Analysis

In order to verify the performance of the complete PDR positioning system combined of improved step estimation, adaptive step length model, and fusion filtering, several sets of pedestrian positioning experiments are carried out indoors.

Multiple tests of pedestrian positioning are performed under status of walking and running in the presented location, respectively. The measured data is stored and then processed evenly to obtain the following simulation map, where the starting point is preset to (0, 0) in matlab. Figure 10 and Figure 11 are actual trajectories and error comparisons in the walking state; Actual trajectories and error comparison diagrams in the running state are shown in Figure 12 and Figure 13. For the quantitative analysis of the improved PDR positioning system, the error range, mean absolute error, and root mean square error of the east and north directions of each trajectory are calculated, as shown in Table 2 and Table 3.

According to the trajectory map under walking, the pedestrian’s trajectory is slowly offset from the real trajectory due to the existence of the cumulative error, and the end point does not coincide with the preset end point. With fusion filter composed of CF and RAKF, the east and north error ranges are −9.8–6.3 m and −5.2–5.1 m, respectively. The mean absolute errors of east and north are 5.3 and 3.8 m. The error analysis with complementary filter and CF + KF are also shown in Table 2. In contrast, the positioning accuracy of fusion filter composed of CF and RAKF is obviously better. The error of the improved PDR positioning system is lower than 2.5% of the total length under walking, which can be well used to conduct accurate positioning in the closed environment.

According to the above figures and Table 3, the positioning accuracy of fusion filter is significantly higher than single complementary filter and CF + KF, which is consistent with the results in the walking state.

### 3.4. Additional Experiments

In this section, in order to verify the scope of the proposed algorithm, the additional experiments in typical sites are performed using two kinds of sensors for testing and comparison. As shown in Figure 14, one is the high-performance sensor Mti series MEMS-IMU from Xsens, and the other is the low-cost IMU AH-100B. The typical sites are divided into complex and broad place.

Typical case 1: The complex site is selected in the laboratory, and the schematic diagram of the laboratory is shown in Figure 15a. Figure 15b displays the setting of experiment: The shadows represent the desks, the blanks represent the walkways, where subject moves. The red circle represents starting point, which is established as the point [0, 0], and the subject moves in the laboratory along the direction of the arrow. The experimental results of Mti series MEMS-IMU are shown in Figure 15c, and Figure 15d represents the positioning trajectory of AH-100B. According to the two figures, it is obvious that the proposed algorithm has better performance than the original model. At the same time, according to Figure 15d, the positioning accuracy of the improved algorithm is significantly improved compared with the traditional algorithm, but it is limited to the performance of the sensor to some extent.

Typical case 2: The wide experimental site is selected in the school library, and the schematic diagram of the library is shown below. The experimental results of Mti series MEMS-IMU are shown in Figure 16b, and Figure 16c represents the positioning trajectory of AH-100B.

According to Figure 16, the experimental results performed in library are consistent with the results in laboratory. In some places where the requirements are not demanding, low-cost MEMS are recommended to be used, whose performance is reluctantly acceptable. However, in some demanding places, such as mines and fire rescue, it is recommended to use high-performance MEMS, which can provide accurate positioning using improved model proposed in this paper.

## 4. Conclusions

In this paper, an improved PDR positioning system based on pure inertial sensors is proposed. First of all, thanks to the development of MEMS-IMU, portable sensors can be carried around. In view of the low signal-to-noise ratio of inertial sensors, wavelet decomposition is used to pre-filter the inertial signals to improve the credibility of the data. In the PDR algorithm, the step estimation and the dynamic step length model are improved to obtain pedestrian’s high precision step and step length. The heading estimation is the core of the PDR algorithm, whose subtle accumulation will seriously affect the accuracy of pedestrian positioning. In heading estimation algorithm, the complementary algorithm is firstly used to fuse magnetometer, accelerometer, and gyroscope to suppress the accumulated errors. Then robust adaptive Kalman filter algorithm is proposed, which can identify and suppress the outliers to improve the stability and accuracy of the positioning system. Finally, the experimental platform with inertial sensors as is established, and the experimental results demonstrate the feasibility and effectivity of the proposed PDR positioning system. Future research will be conducted to study the body kinematics model to improve the PDR algorithm. In addition, the accuracy of the PDR-based algorithm is closely related to the location of the MEMS-IMU is the focus of future research.

## Figures and Tables

**Figure 1 sensors-19-00294-f001:**
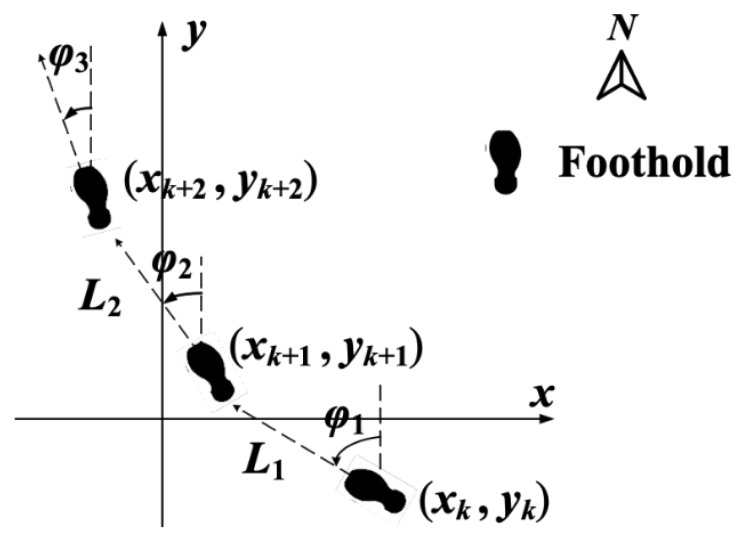
Pedestrian dead reckoning model.

**Figure 2 sensors-19-00294-f002:**
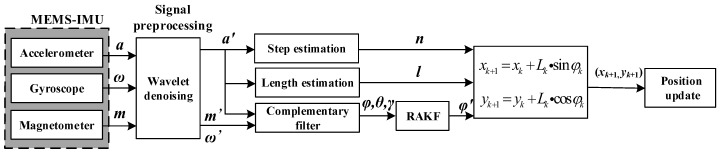
Pedestrian positioning system schematic. MEMS-IMU: microelectromechanical-inertial measurement unit; RAKF: robust adaptive Kalman filter.

**Figure 3 sensors-19-00294-f003:**
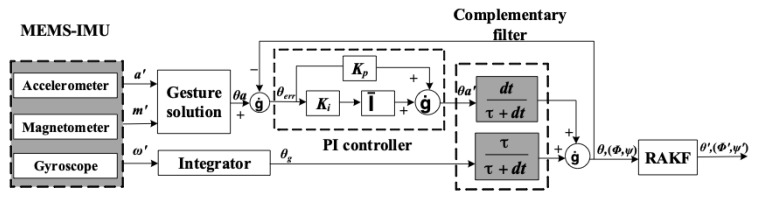
Fusion filter model.

**Figure 4 sensors-19-00294-f004:**
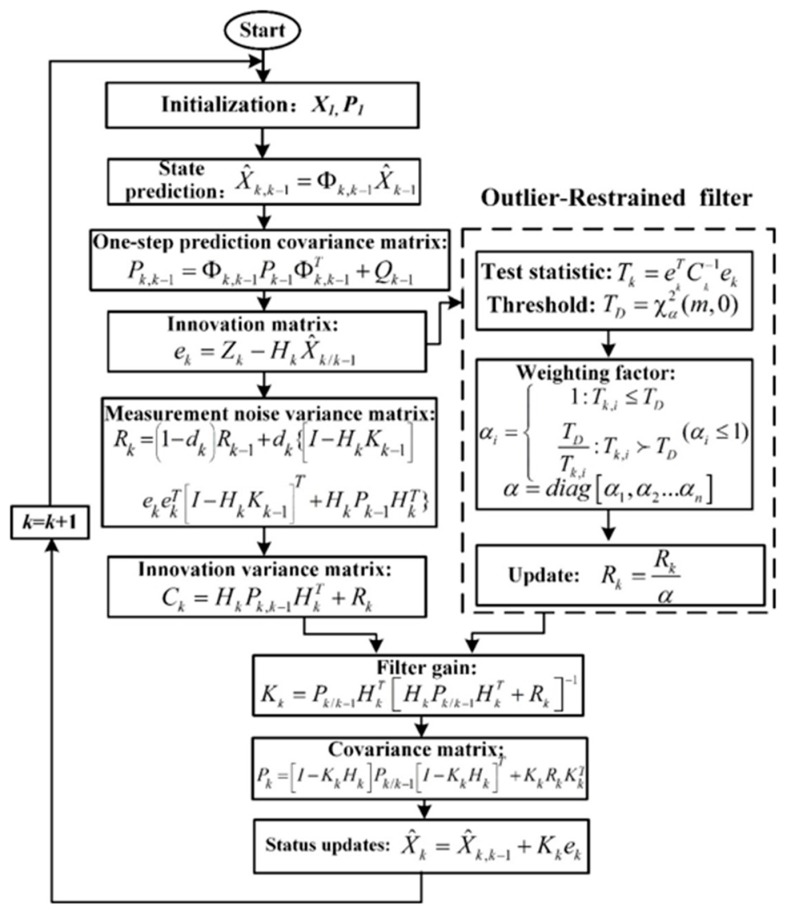
RAKF algorithm.

**Figure 5 sensors-19-00294-f005:**
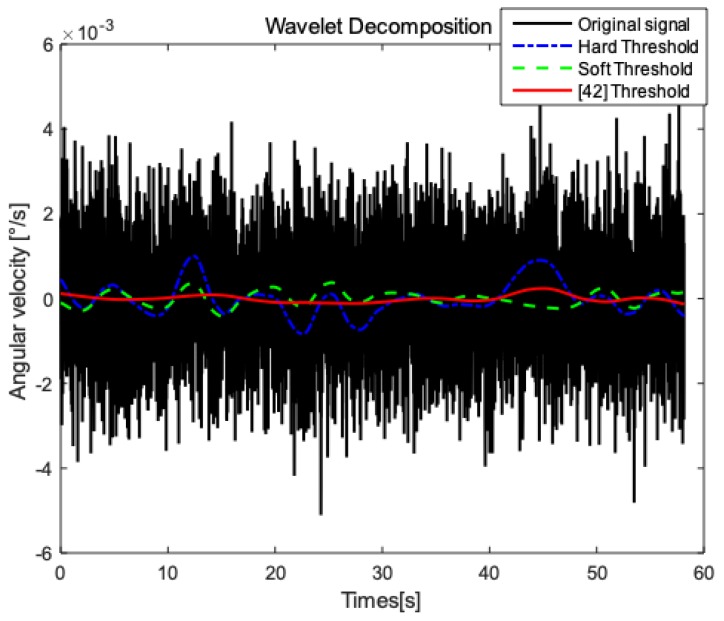
Comparison of original signal and reconstructed signals based on different thresholds.

**Figure 6 sensors-19-00294-f006:**
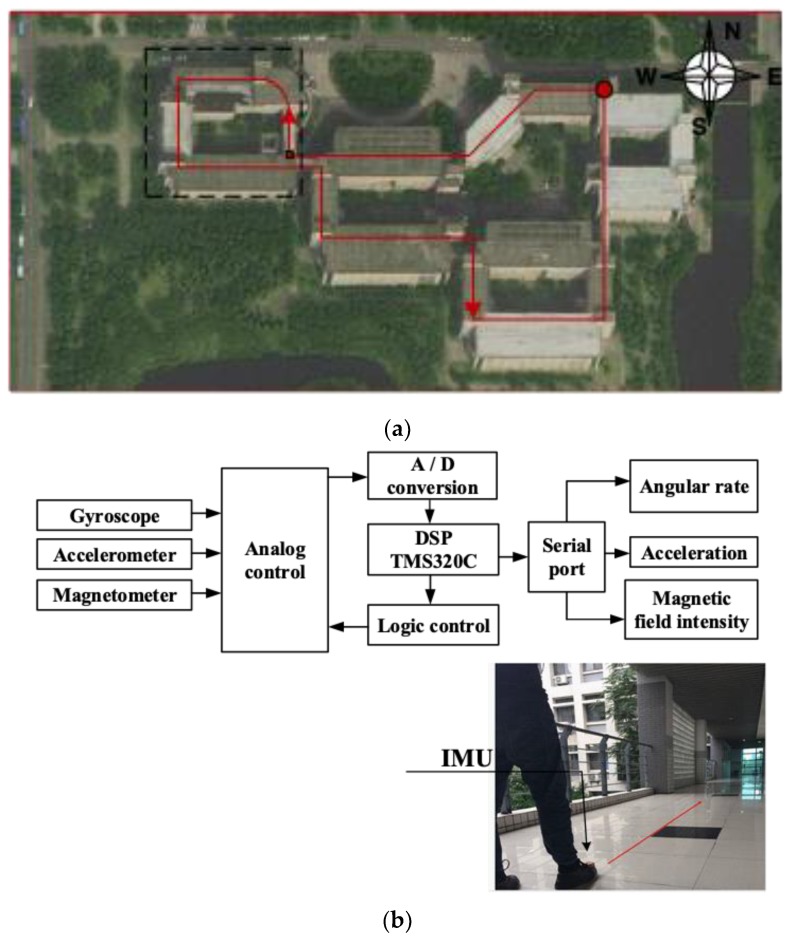
(**a**) Satellite image of the experimental site (**b**). Inertial measurement unit (IMU) structure and installation diagram.

**Figure 7 sensors-19-00294-f007:**
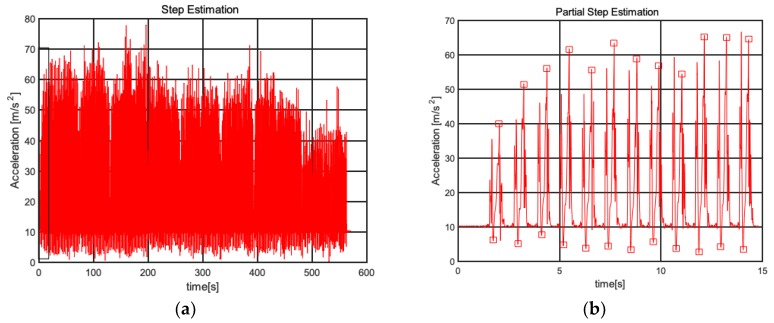
Step estimation. (**a**) Simulation diagram of step detection; (**b**) Partial magnification simulation.

**Figure 8 sensors-19-00294-f008:**
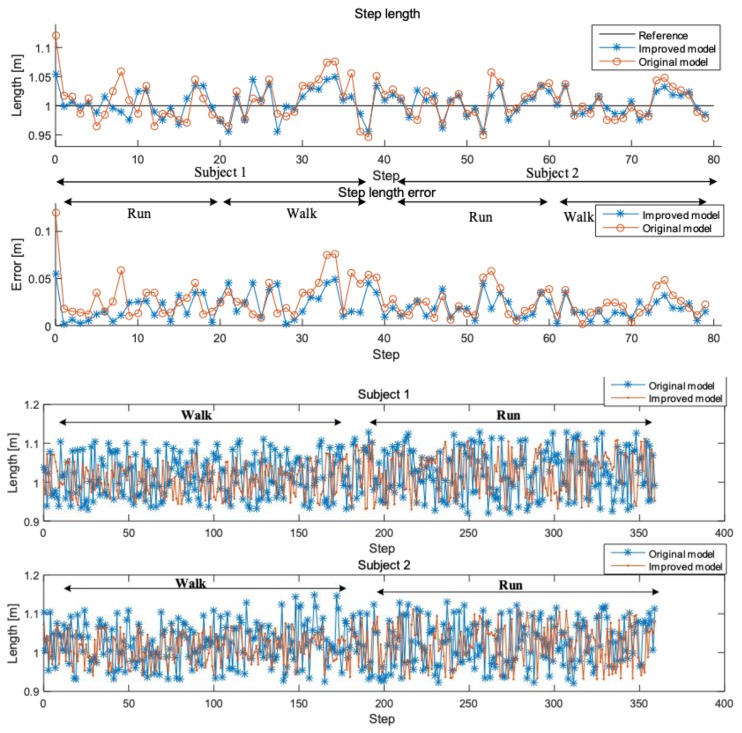
Step length and error comparison of two subjects in different motions using different models.

**Figure 9 sensors-19-00294-f009:**
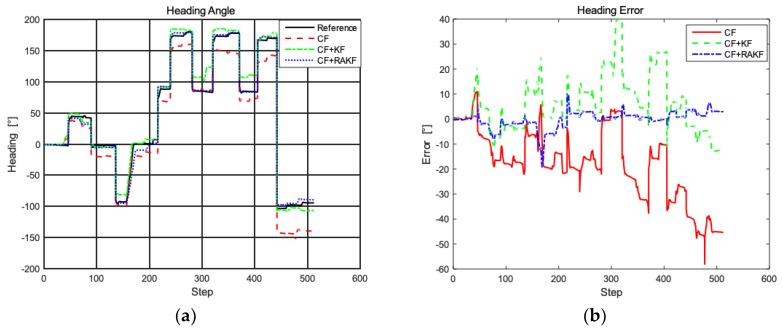
Heading analysis in different filtering strategies. (**a**) Comparison of heading; (**b**) Heading error comparison.

**Figure 10 sensors-19-00294-f010:**
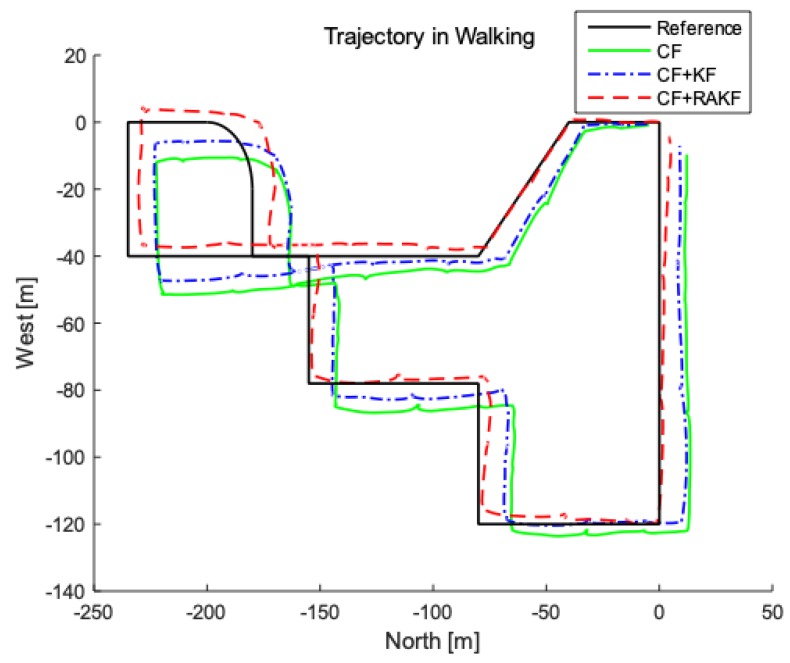
Comparison of trajectory in walking under different heading estimation algorithms performed in second floor of entire Internet of Things (IOT) Engineering College.

**Figure 11 sensors-19-00294-f011:**
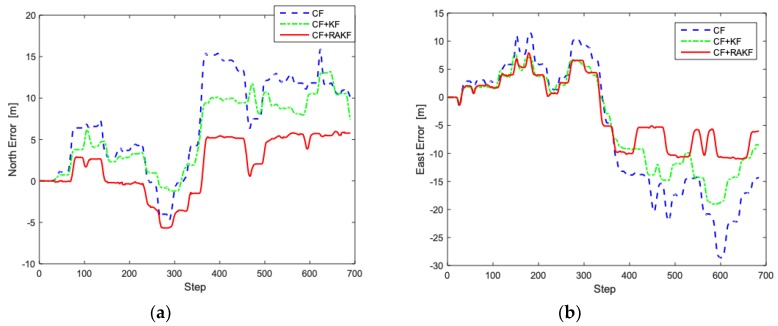
Trajectory analysis in walking under different heading estimation algorithms. (**a**) North error comparison; (**b**) East error comparison.

**Figure 12 sensors-19-00294-f012:**
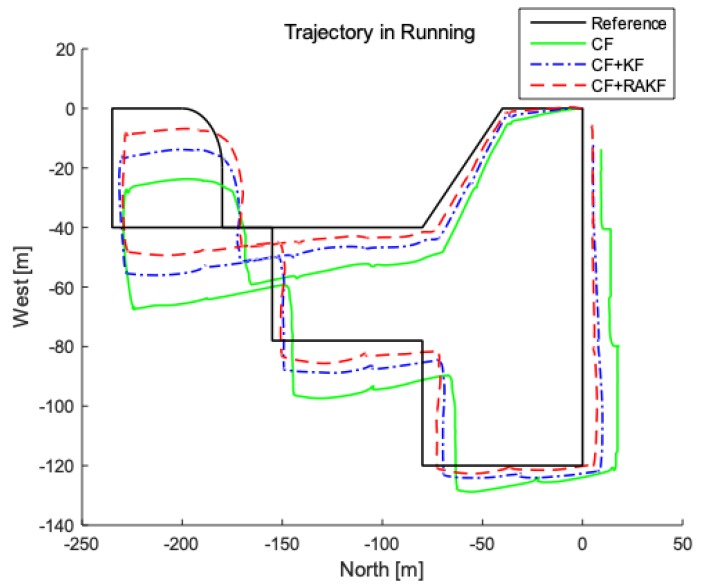
Comparison of trajectory in running under different heading estimation algorithms performed in second floor of IOT Engineering College.

**Figure 13 sensors-19-00294-f013:**
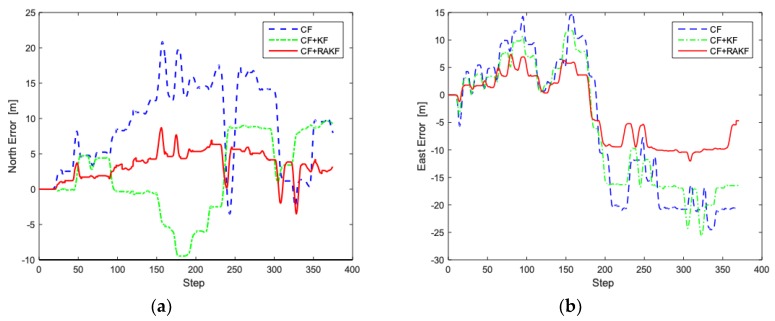
Trajectory analysis in running under different heading estimation algorithms. (**a**) North error comparison; (**b**) East error comparison.

**Figure 14 sensors-19-00294-f014:**
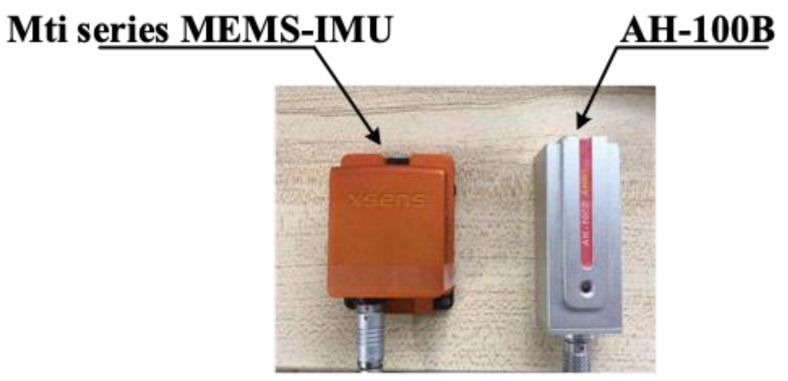
Experiment apparatus, Mti series MEMS-IMU from Xsens and AH-100B.

**Figure 15 sensors-19-00294-f015:**
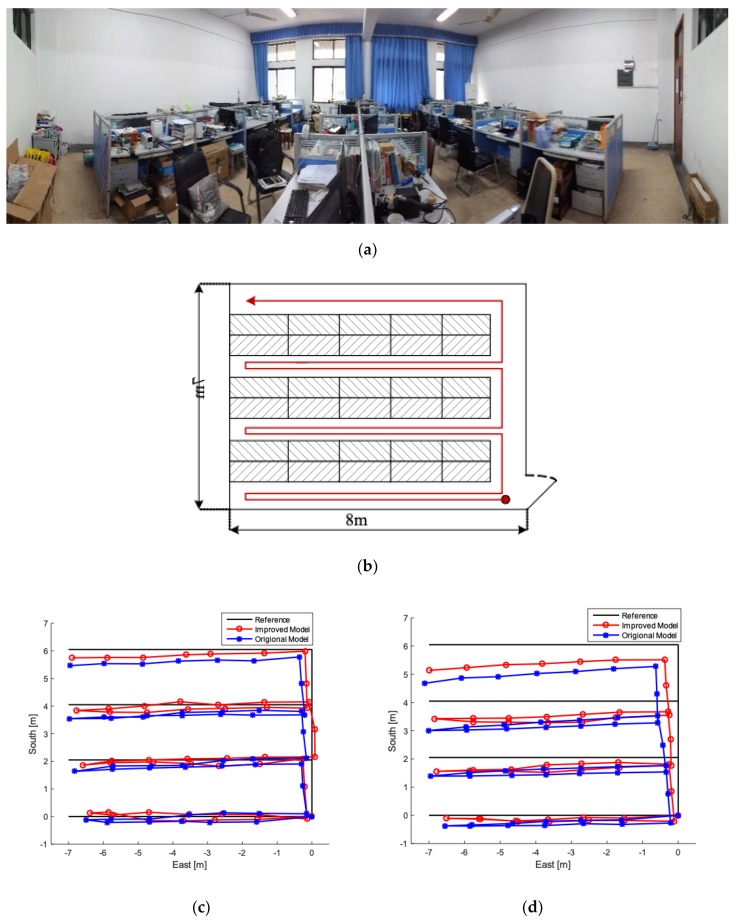
Typical experiment 1. (**a**) Laboratory under panoramic view; (**b**) experimental diagram. (**c**) experimental results of Mti series MEMS-IMU from Xsens. (**d**) Experimental results of AH-100B.

**Figure 16 sensors-19-00294-f016:**
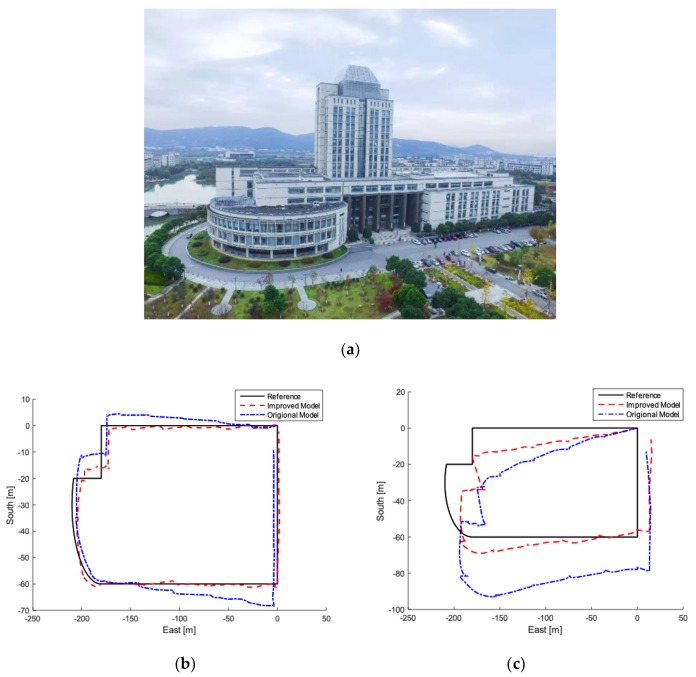
Typical experiment 2. (**a**) Experimental site; (**b**) Experimental results of Mti series MEMS-IMU from Xsens. (**c**) Experimental results of AH-100B.

**Table 1 sensors-19-00294-t001:** Step length analysis of two subjects using different models. RD: realistic distance; SD: solution distance; MAE: mean absolute error.

	RD [m]	Original Model		Improved Model	
		SD [m]	MAE [m]	SD [m]	MAE [m]
**Run-1**	181.0	195.1	0.13	189.3	0.08
**Walk-1**	181.0	174.3	0.09	175.6	0.07
**Run-2**	181.0	200.9	0.15	191.5	0.11
**Walk-2**	181.0	207.5	0.18	187.1	0.12

**Table 2 sensors-19-00294-t002:** Error comparison table in walking under different heading estimation algorithm. CF: complementary filtering; KF: Kalman filter.

	CF		CF + KF		CF + RAKF	
	North	East	North	East	North	East
Error Range [m]	−4.8–15.1	−27.5–12.3	−2.7–14.6	−16.3–6.7	−5.2–5.1	−9.8–6.3
MAE [m]	10.7	14.7	8.1	9.4	3.8	5.3

**Table 3 sensors-19-00294-t003:** Error comparison table in running under different heading estimation algorithm.

	CF		CF + KF		CF + RAKF	
	North	East	North	East	North	East
Error Range [m]	−4.7–21.3	−25.1–12.6	−8.5–10.3	−25.3–11.2	−4.6–8.0	−12.9–7.4
MAE [m]	11.3	16.9	8.3	13.2	4.7	7.2

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
