# Peer review of "Improved Pedestrian Dead Reckoning Based on a Robust Adaptive Kalman Filter for Indoor Inertial Location System"

_sensors, 2019, doi:10.3390/s19020294_

Round 1
Reviewer 1 Report
The topic is interesting. Although the manuscript is well-written, some issues have to be addressed in order to improve its clarity and applicability.
In introduction, it is said that UWB, ZigBee, Wi-Fi, Bluetooth, RFID, and virtual positioning are used for indoor positioning and are easily interfered. A discussion for UWB is presented, but not for the other technologies, which is important in order to show the advantages of the proposal. I
n page 2, it is said that MEMS-IMU presents a set of advantages, but they are not mentioned. Wavelet-based pre-processing is an important step in the proposal. In this regard, some points have to be clarified. How were the wavelet basis function and the number of decomposition layers selected? Which is the sampling frequency? For a lower sampling frequency, will a lower decomposition level be suitable?
A more detailed justification for the selection of 0.7 and 1.2 (complementary filter), and 0.98 and 0.1 (Kalman filter) values has to be included. Results for CF and CF+RAKF are compared. A comparison with the traditional Kalman (page 7 of 18) is needed in order to show the need of the RAKF.
What do you mean with sufficient tests in section 3.3.2? Is a new training data set required for a new subject? For specific values of alfa and beta, how is the behavior of the proposal if a new training is not carried out?
Figure 4 is not mentioned in the manuscript.
Figure 5 shows both a noisy signal and a denoised signal. From this figure, it is said that results prove the filtering effectiveness, however there is not a reference signal. How can the Authors ensure the filtering effectiveness? A synthetic signal with known parameters can be used for this test, even including other denoising signal processing techniques.
In Figure 6, a picture of the MEMS-IMU-based system is desirable.
Errors in Figure 8 appear to be very similar. Is there a really improvement? Which is the computational cost of the proposal? Is it applied online and real-time? Are the data stored or transmitted and then processed? Which is the platform used for processing? Information about the implementation of the proposal is barely described.
Which is the requirement of high precision positioning in pedestrians (page 12 of 18)?
How was the confidence computed?
Please describe all the acronyms,
References have to be presented in a consecutive order, please check them.
Author Response
Response to Reviewer 1 Comments
Dear editor and reviewers:
Thank you very much for your advice on our manuscript. We have resubmitted new version of manuscript in accordance with recommendations of you. In this revised version, changes to our manuscript were all highlighted by using red text in revision mode. Point-by-point responses to reviewer 1 are listed below this letter.
Point 1: In introduction, it is said that UWB, ZigBee, Wi-Fi, Bluetooth, RFID, and virtual positioning are used for indoor positioning and are easily interfered. A discussion for UWB is presented, but not for the other technologies, which is important in order to show the advantages of the proposal. In page 2, it is said that MEMS-IMU presents a set of advantages, but they are not mentioned. Wavelet-based pre-processing is an important step in the proposal. In this regard, some points have to be clarified. How were the wavelet basis function and the number of decomposition layers selected? Which is the sampling frequency? For a lower sampling frequency, will a lower decomposition level be suitable?
Response 1: Thank you for your suggestion, which will improve the quality of my paper. The discussions for the other technologies is added in my revised version. In page 2, a set of advantages of MEMS-IMU are not mentioned, and we have added as shown in revised version. The choice of wavelet basis function is mainly based on the mathematical properties, such as orthogonality and regularity. The 'db6' selected in this paper mainly refers to reference [39]. The number of decomposition layers of 3 to 9 wavelets was selected for comparative study, and the optimal decomposition layer number was selected. the sampling frequency is 100hz, and the current number of levels is sufficient.
Point 2: A more detailed justification for the selection of 0.7 and 1.2 (complementary filter), and 0.98 and 0.1 (Kalman filter) values has to be included. Results for CF and CF+RAKF are compared. A comparison with the traditional Kalman (page 7 of 18) is needed in order to show the need of the RAKF.
Response 2: In complementary filter, parameters are selected according to reference [41]. Kp is the switching frequency of the low-pass filter and the high-pass filter, which is selected according to the frequency characteristics of the accelerometer and the output of the gyroscope, and Kp = Ki2/2. In the literature [42], the range of b is 0.95-0.99, and we make b=0.98 in the paper. The significance level α indicates the probability that the null hypothesis is established but rejects the hypothesis selection alternative hypothesis. The selection of the significance level determines the threshold of the innovation statistic. The smaller the value of α, the larger the threshold, and too large or too small threshold will affect the system's robust performance. A suitable value for the alpha should be in the range of 0.01 to 0.15, and we make α=0.1 in the paper. Finally, we are sorry for our negligence of something important as your suggestion. A comparison with the traditional Kalman is added in Figure 9-13 and Table 2-3 to prove the effectiveness of RAKF.
Point 3: What do you mean with sufficient tests in section 3.3.2? Is a new training data set required for a new subject? For specific values of alfa and beta, how is the behavior of the proposal if a new training is not carried out?
Response 3: Please let me express apologies for my unclear expression firstly. Sufficient tests in section 3.3.2 mean that we shall collect as much data as possible to perform the least squares method to determine the parameters α and β. A new training data set is required for a new subject. If alfa and beta are specific for different individuals, the behaviour of the proposal may be worse than original model, when the difference between individuals is too large, such as large difference between height, weight.
Point 4: Figure 4 is not mentioned in the manuscript.
Response 4: Thank you for pointing out the question, the appearance of Figure 4 is indeed too abrupt. We have I have added the introduction of the Figure.
Point 5: Figure 5 shows both a noisy signal and a denoised signal. From this figure, it is said that results prove the filtering effectiveness, however there is not a reference signal. How can the Authors ensure the filtering effectiveness? A synthetic signal with known parameters can be used for this test, even including other denoising signal processing techniques.
Response 5: We are sorry for our negligence of something important as your suggestion. In Figure 5, the denoised signals in hard and soft threshold functions are added to prove the filtering effectiveness.
Point 6: In Figure 6, a picture of the MEMS-IMU-based system is desirable.
Response 6: Thank you for pointing this out, and I have added the MEMS-IMU-based system.
Point 7: Errors in Figure 8 appear to be very similar. Is there a really improvement? Which is the computational cost of the proposal? Is it applied online and real-time? Are the data stored or transmitted and then processed? Which is the platform used for processing? Information about the implementation of the proposal is barely described.
Response 7: Thank you for pointing this out, please allow me to explain to you. Figure 8 shows comparison of different models. The comparison between the different models is not obvious because single step is relatively short. But after the cumulative effect, the gap between the models is more obvious according to Table 1. The computational cost is. The system is offline, and computational cost is low. Data is stored and then processed in matlab, which I have supplemented in my paper.
Point 8: Which is the requirement of high precision positioning in pedestrians (page 12 of 18)?
Response 8: Please let me express apologies firstly for my unclear expression. The requirement of high precision positioning in pedestrians means that the improved length estimation model can provide accurate length estimation in PDR algorithm, which has been modified as shown revised version.
Point 9: How was the confidence computed?
Response 9: Please let me express my apologies firstly, it is a giant mistake to the quality of our article. The formula of confidence is shown as follows:
Where n represents the number of data, xi represents the i-th measurement data, and x represents the reference data.
In my previous calculations, the data whose reference value is 0 was discarded. But I now find that confidence does not make sense here because the confidence is affected by the reference value, which is affected by the set of starting point. For the same graph, setting different starting points will result in different confidence levels, which is obviously wrong. So I gave up the confidence here.
Point 10: Please describe all the acronyms.
Response 10: Thank you for your suggestion, which will improve the quality of my paper. I have described all the acronyms, including GPS, UWB and so on.
Point 11: References have to be presented in a consecutive order, please check them.
Response 11: Thank you for pointing this out, I have checked all the references and present them in a consecutive order
We appreciate for your comments, and hope that the response and correction will meet with approval.
Yours sincerely.
Hai Zhang
2018.12.26
Reviewer 2 Report
MEMS-based PDR is not a new topic.
The designed trajectories for your research are too simple, I recommend the authors to test in a more flexible trajectory.
The sensor drift error of low-cost MEMS is large, and it is very sensitive to the environment and operating time, the wavelet denoising algorithm is not that effective for this situation.
What is the definition of "high-precision positioning"
The author mentioned that the error is 2% of the total distance, however, the testing distance is too short, not that convincing.
Mti series MEMS-IMU from Xsens is not a low-cost IMU, it is not commonly used for commercial users. The measurement outputs are much better than the dollar-level IMU in a portable device, I recommend the author to test in a dollar level IMU.
Author Response
Response to Reviewer 2 Comments
Dear editor and reviewers:
Thank you very much for your advice on our manuscript. We have resubmitted new version of manuscript in accordance with recommendations of you. In this revised version, changes to our manuscript were all highlighted by using red text in revision mode. Point-by-point responses to reviewer 2 are listed below this letter.
Point 1: The designed trajectories for your research are too simple, I recommend the authors to test in a more flexible trajectory.
Response 1: Thank you for pointing this out, the previously designed trajectory was simple and not long enough. In the revised manuscript, a more complex, longer trajectory is used for testing.
Point 2: The sensor drift error of low-cost MEMS is large, and it is very sensitive to the environment and operating time, the wavelet denoising algorithm is not that effective for this situation.
Response 2: Thank you for pointing out the question, please allow me to explain to you. The sensor selected in this paper is Mti series MEMS-IMU from Xsens, which does not belong to low-cost IMU. The performance of Mti-MEMS-IMU is better than low-cost IMU, and the wavelet denoising algorithm is effective in our research.
Point 3: What is the definition of "high-precision positioning"
Response 3: Thank you for your comment, and show my sincere respect for your rigorous academic attitude firstly. The use of "high-precision positioning" is too casual, and has been changed into “accurate”. And the definition of "accurate" is that the positioning accuracy of new system is better than traditional inertial positioning system.
Point 4: The author mentioned that the error is 2% of the total distance, however, the testing distance is too short, not that convincing.
Response 4: Thank you for your comment. The previous test distance is not long enough, where the result is not convincing. In the revised manuscript, a longer trajectory is used for testing, which can make results convincing.
Point 5: Mti series MEMS-IMU from Xsens is not a low-cost IMU, it is not commonly used for commercial users. The measurement outputs are much better than the dollar-level IMU in a portable device, I recommend the author to test in a dollar level IMU.
Response 5: Thank you for pointing this out, please allow me to explain to you. As expert say, Xsens' Mti series MEMS-IMUs are not low-cost IMUs, and the measurement output is much better than the dollar-level IMU. But there is still a large error when using Mti series MEMS-IMU to perform positioning experiments, which can be improved. In addition, Mti series MEMS-IMU from Xsens is not indeed suitable for commercial use because of its expensive price. However, this paper is set in the context of emergency rescue, such as fire and mine rescue, and use pure inertial navigation for positioning. So accuracy is our focus, not whether it can be commonly used for commercial users.
We appreciate for your comments, and hope that the response and correction will meet with approval.
Yours sincerely.
Hai Zhang
2018.12.26
Reviewer 3 Report
This paper presents an improved pedestrian dead reckoning method using robust adaptive Kalman filter. This topic is interesting and important, there are some issues that require to be addressed.
The novelty is unclear. Have the authors improved all the three components of PDR? If yes, it should be highlighted in the introduction part, and must be supported with extensive experiments.
Each symbol and sign in equations has to be clearly explained, e.g., which is T, and u in equation (2).
In equations (4), (5), where pn and vn should be larger than 10? It needs more explanations.
Have you considered different device poses? (e.g., in the pocket, in hand).
The test distance for step length evaluation is too short (only 20 meters). Multiple paths of different different length (e.g., 100m) should be considered.
What is the difference between the fusion filter and CF+RAKF? It appears in different figures, which is confusing.
What is confidence? It needs to be defined or explained before using.
What is PDR 1 and PDR 2? Is it PDR for different sites?
Author Response
Response to Reviewer 3 Comments
Dear editor and reviewers:
Thank you very much for your advice on our manuscript. We have resubmitted new version of manuscript in accordance with recommendations of you. In this revised version, changes to our manuscript were all highlighted by using red text in revision mode. Point-by-point responses to reviewer 3 are listed below this letter.
Point 1: The novelty is unclear. Have the authors improved all the three components of PDR? If yes, it should be highlighted in the introduction part, and must be supported with extensive experiments.
Response 1: Thank you for pointing out the question. In the PDR algorithm, step estimation and heading estimation are mainly improved. And the improvements have been highlighted in introduction as shown in the revised version. Experimental trajectory is improved to be more complex and longer, and extensive experiments are tested in the experimental part.
Point 2: Each symbol and sign in equations has to be clearly explained, e.g., which is T, and u in equation (2).
Response 2: We are sorry for our negligence of something important as your suggestion, which make readers confused. T is the denoising operation, that is, by the threshold and the threshold function, filtering out the noise coefficient and retaining the detail coefficient. u is the wavelet coefficient before denoising. The supplementary part has been added in the revised version.
Point 3: In equations (4), (5), where pn and vn should be larger than 10? It needs more explanations
Response 3: We are sorry for our negligence to explain something important. When stationary, the accelerometer is affected by the acceleration of gravity, and the total acceleration A is approximately equal to 10, as shown in Figure 7. In the cycle of foot movement, starting with the footstep off the ground and ending with the footstep touching the ground again. At the start, the upward acceleration is opposite to the direction of gravity acceleration, and the direction is the same when re-contacting, so set pη>10 and vη<10.
Point 4: Have you considered different device poses? (e.g., in the pocket, in hand).
Response 4: Thank you for your comment. At present, our device is mounted and fixed on the instep. In this case, the step estimation we proposed is effective. Therefore, there is no choice of other installation parts or methods.
Point 5: The test distance for step length evaluation is too short (only 20 meters). Multiple paths of different length (e.g., 100m) should be considered.
Response 5: Thank you for your pertinent comment firstly. The previous test distance for step length evaluation is indeed not long enough, where the result is not convincing. In the revised manuscript, a longer and more complex trajectory is used for testing, which can make results convincing. The specific experimental path for step length evaluation is shown in Figure 6(a).
Point 6: What is the difference between the fusion filter and CF+RAKF? It appears in different figures, which is confusing.
Response 6: Please let me express my apologies firstly. It is my fault that didn't use a uniform format, which may make you and other readers confusing. Although fusion filter and CF+RAKF are the same in the paper, CF+RAKF is uniformly used in all figures. We can see all the changes in revised manuscript.
Point 7: What is confidence? It needs to be defined or explained before using.
Response 7: Please let me express my apologies firstly, it is a giant mistake to the quality of our article. The formula of confidence is calculated as follows:
Where n represents the number of data, xi represents the i-th measurement data, and x represents the reference data.
In my previous calculations, the data whose reference value is 0 was discarded. But I now find that confidence does not make sense here because the confidence is affected by the reference value, which is affected by the set of starting point. For the same graph, setting different starting points will result in different confidence levels, which is obviously wrong. So I gave up the confidence here.
Point 8: What is PDR 1 and PDR 2? Is it PDR for different sites?
Response 8: Please let me express apologies firstly for my unclear expression. Pedestrian performs multiple experiments at the experimental sites, and the improved PDR algorithms is used to obtain the localization maps. Among them, two localization maps of each site are selected and displayed in the Figure 14 and Figure 15. So PDR1 and PDR2 are trajectories based on improved PDR location system. The specific explanation has been added in the revised manuscript.
We appreciate for your comments, and hope that the response and correction will meet with approval.
Yours sincerely.
Hai Zhang
2018.12.27
Round 2
Reviewer 1 Report
All my concerns have been addressed. I consider that the manuscript is suitable for publication. My last suggestion is to include and discuss information of response 3 in the manuscript.
Author Response
Response to Reviewer 1 Comments
Dear editor and reviewers:
Thanks very much for your advice on our manuscript. We have resubmitted new version of manuscript in accordance with recommendations of you. In this revised version, changes to our manuscript were all highlighted, which are different from changes in round 1. Point-by-point responses to reviewer 1 are listed below this letter.
Point 1: All my concerns have been addressed. I consider that the manuscript is suitable for publication. My last suggestion is to include and discuss information of response 3 in the manuscript.
Response 1: First of all, thanks for your suggestions and your affirmation of our work. Then, for specific values of alfa and beta, the performance of the proposal algorithm is poor, which is even worse than original model with corresponding parameters. We have discussed the performance of specific values in new version of manuscript, which are all highlighted.
We appreciate for your comment and affirmation, and hope that the response and correction will meet with approval.
Yours sincerely.
Hai Zhang
2019.1.5
Reviewer 2 Report
C 1. PDR method is a mature research method for inertial navigation area, however, for other general readers it might be obscure to understand, it is better to cite some publications. Followed by are good examples, you can cite directly or you can find some other publications.
1. Yu, C., Lan, H., Gu, F., Yu, F., & El-Sheimy, N. (2017). A Map/INS/Wi-Fi Integrated System for Indoor Location-Based Service Applications. Sensors, 17(6), 1272.
2. Chiang, K. W., Liao, J. K., Tsai, G. J., & Chang, H. W. (2016). The performance analysis of the map-aided fuzzy decision tree based on the pedestrian dead reckoning algorithm in an indoor environment. Sensors, 16(1), 34.
3. Zampella, F., De Angelis, A., Skog, I., Zachariah, D., & Jimenez, A. (2012, November). A constraint approach for UWB and PDR fusion. In Indoor Positioning and Indoor Navigation (IPIN), 2012 International Conference (pp. 1-9). IEEE.
C 2. Can you test and compare the proposed algorithm on Xsens and the dollar-level MEMS sensors, even the first response force, Xsens is not a possible choice because of its high cost, for engineering application, cost and accuracy need to be trade-off all the time.
Author Response
Response to Reviewer 2 Comments
Dear editor and reviewers:
Thanks very much for your advice on our manuscript. We have resubmitted new version of manuscript in accordance with recommendations of you. In this revised version, changes to our manuscript were all highlighted, which are different from changes in round 1. Point-by-point responses to reviewer 2 are listed below this letter.
Point 1: PDR method is a mature research method for inertial navigation area, however, for other general readers it might be obscure to understand, it is better to cite some publications. Followed by are good examples, you can cite directly or you can find some other publications.
Response 1: Thank you for pointing this out, which is indeed my negligence. I have read the literatures you have recommended, and these documents can introduce PDR method well, which helps the general reader understand PDR. These documents have been added in this revised version, and the serial number of the documents has been readjusted.
Point 2: Can you test and compare the proposed algorithm on Xsens and the dollar-level MEMS sensors, even the first response force, Xsens is not a possible choice because of its high cost, for engineering application, cost and accuracy need to be trade-off all the time.
Response 2: Thank you for pointing out the question. As recommended by the expert, the additional experiments are performed using two kinds of sensors for testing and comparison. one is the high-performance sensor Mti series MEMS-IMU from Xsens, and the other is the low-cost IMU AH-100B, which are shown in Figure 14.
The experimental results are shown in Figures 15 and 16. According to the figures, it is obvious that the proposed algorithm has better performance than the original model. At the same time, the positioning accuracy of the improved algorithm is significantly improved compared with the traditional algorithm, but it is also limited to the performance of the sensor to some extent. Therefore, in some places where distance is short and requirements are not demanding, low-cost MEMS is recommended to be used because of its low cost, whose performance can be reluctantly acceptable using improved model proposed in this paper. However, in some demanding places, such as mines and fire rescue, it is recommended to use high-performance MEMS, which can provide accurate positioning using improved model proposed.
We appreciate for your comments, and hope that the response and correction will meet with approval.
Yours sincerely.
Hai Zhang
2019.1.5
Reviewer 3 Report
The authors have made some significant revisions according to the comments, which are appreciated. There are minor comments that should be addressed to further improve the quality of this paper.
This manuscript should be proofread by native speakers or professionals, some expressions are unsuitable, E.g., Literature [40] (in 2.3.2) is not suitable, which can be changed as "The authors in [40] use ..."
The trajectories in the figures should be adjusted to be a bit bolder for clarity.
Author Response
Response to Reviewer 3 Comments
Dear editor and reviewers:
Thanks very much for your advice on our manuscript. We have resubmitted new version of manuscript in accordance with recommendations of you. In this revised version, changes to our manuscript were all highlighted, which are different from changes in round 1. Point-by-point responses to reviewer 3 are listed below this letter.
Point 1: This manuscript should be proofread by native speakers or professionals, some expressions are unsuitable, E.g., Literature [40] (in 2.3.2) is not suitable, which can be changed as "The authors in [40] use ...".
Response 1: Thank you for pointing out the question. The unsuitable expression you proposed has been modified, and we invite English teacher to proofread this manuscript. All changes to our manuscript were all highlighted in this revised version, which are different from changes in round 1.
Point 2: The trajectories in the figures should be adjusted to be a bit bolder for clarity.
Response 2: Thank you for pointing out the question, the trajectories in the figures are indeed thin, which are hard for clarity. We have adjusted the trajectories to be a bit bolder, and replace the previous figures. In order to reduce the size of the text, we directly discarded the previous figures and only retained the modified figures. The supplementary part has been added in the revised version. The modification in round 2 is different in colour from changes in round 1 in this revised version.
Lastly, thanks for your suggestions and your affirmation of our work, and we hope that the response and correction will meet with approval.
Yours sincerely.
Hai Zhang
2019.1.5
Round 3
Reviewer 2 Report
Thanks for your reply